

# The semi-annual oscillation (SAO) in the upper troposphere and lower stratosphere (UTLS)

Ming Shangguan[1] and Wuke Wang[2]

[1]School of Geography and Informaiton Engineering,China University of Geosciences,Wuhan,China
[2]Department of Atmospheric Science,China University of Geosciences,Wuhan,China

**Correspondence:** Wuke Wang (wangwuke@cug.edu.cn)

**Abstract.** Both the scientific and operational communities are increasingly interested in subseasonal to seasonal variations of weather and climate. The Semi-Annual Oscillation (SAO) has been studied extensively at the surface as well as in the middle atmosphere (upper stratosphere and the lower mesosphere). However, the SAO in the upper troposphere and lower stratosphere (UTLS) has been less discussed. Here we find evident SAO of temperature in the UTLS (250-175 hPa) from the subtropics

to mid-latitudes (22.5-42.5°) using high-quality satellite measurements, reanalysis data, and model simulations. We show the mechanism of its formation by an energy budget analysis. The temperature in the Northern Hemisphere (NH) UTLS shows the first peak in February according to the dynamical heating and shows the second peak in July due to the dynamical heating and moist processes. Similar to the NH, the winter time maximum of temperature in the SH is related to dynamical heating and the summer time maximum is related to both moist and dynamical heating in the UTLS. Model simulations indicate that the SAO

in the UTLS is partly modified by the SAO of sea surface temperatures (SSTs).

## 1 Introduction

Subseasonal to seasonal predictions of weather and climate are increasingly important due to the urgent requirement from decision makers (Merryfield et al., 2020). As an important component of subseasonal to seasonal variations, the Semi-Annual Oscillation (SAO) has been well known at the surface (surface-SAO) (Meehl et al., 1998) as well as in the middle atmosphere

(MA-SAO) (Garcia et al., 1997). It is well known that the annual cycle of temperature gradient and mean sea level pressure is dominated by a strong half-yearly oscillation between 50°S and 65°S, which exceeds the magnitude of the yearly wave in many locations in the SH near the surface (Walland and Simmonds, 1998; Simmonds and Jones, 1998). This surface-SAO is a coupled ocean-atmosphere phenomenon, and a change of the seasonal cycle of Sea Surface Temperatures (SSTs) at 50°S could alter the amplitude of the surface-SAO (Meehl et al., 1998). The SAO is also noticeable as the dominating mode of wind and

temperature variations in the tropical middle atmosphere between the middle stratosphere and the upper mesosphere (Read, 1962; Garcia et al., 1997). This MA-SAO is driven by the advection of momentum by the mean meridional circulation and the dissipation of vertically propagating equatorial waves (Richter and Garcia, 2006).

In comparison, only a few studies focus on the SAO in the upper troposphere and lower stratosphere (UTLS). The SAO in temperature is reported by Loon (1967) over the middle troposphere in the SH with two maximum in March and September near





the equinoxes, which is related to varying heating/cooling rates in different latitude bands. Loon and Jenne (1969) described the tropical-subtropical SAO of the zonal wind and temperature in the SH based on data of the 20 radiosonde stations in the upper troposphere (100-300 hPa). They also found that the amplitude of SAO in temperature and wind are different along different longitudes, i.e., with stronger signals in the eastern hemisphere (Loon and Jenne, 1969; Chen and Tsay, 2014). According to Loon and Jenne (1969), the temperature oscillation is the result of an intensification of vertical motions from autumn to winter

and the zonal wind oscillation is associated with second harmonics of opposite phase in the temperature. Shea et al. (1995) made a further study on the Tropical-Subtropical Semi-Annual Oscillation (TS-SAO) in the upper troposphere with more extensive data coverage. Shea et al. (1995) found that the TS-SAO of temperature, which has maxima in the transitional seasons in the tropics and peaks in the extreme seasons in the subtropics, resulted from a marked semi-annual variation of the winds in the tropics with maxima in the inter-monsoonal months. The precise mechanism for this oscillation seems somewhat unclear, but

may be associated with thermal fluctuations connected with the movement of the intertropical convergence zone from one side of the Equator to the other and associated modulations of the mean meridional circulation (Read and Castrejón-Pita, 2012).

While measurements in the UTLS are relatively sparse, reanalysis data are widely used to investigate temperature variabilities (Broeke, 2000; Fueglistaler et al., 2009; Gettelman et al., 2011; Wang et al., 2016; Shangguan et al., 2019). Despite the importance of the SAO in the mid-latitudes, there have been few articles focused on it since that of Loon (1967), and these

have been mostly based on radiosonde and reanalysis data. With the development of satellite techniques, many new remote sensing data are available in the UTLS. According to many studies (Wickert et al., 2001, 2009; Ho et al., 2017), the Global Navigation Satellite System Radio Occultation (GNSS RO) can provide highly accurate temperature profiles from the middle-upper troposphere to lower stratosphere by measuring the time delay in occulted signals from one satellite to another. The first RO mission (Challenging Minisatellite Payload (CHAMP)) was launched in 2001, and the Constellation Observing System for

Meteorology, Ionosphere and Climate (COSMIC), which is a constellation of six satellites since late 2006 is widely applied in the climate research (Ho et al., 2014; Randel and Wu, 2015; Gao et al., 2017). Compared with radiosonde data, the GNSS RO data have global coverage (both in continents and oceans) and are not affected by weather (Anthes et al., 2008).

In this study, we use the GNSS RO data and the latest ECMWF reanalysis ERA5 (Hersbach et al., 2019) and the NASA MERRA2 reanalysis (GMAO, 2015a, b) to study the SAO in the UTLS (UTLS-SAO), and to discuss its origin. We expand

the oscillation using temperature data available from GNSS RO and reanalyses between 60°N to 60°S spanning the period 2001-2017, and using Hadley Centre SST data set (Rayner et al., 2003) for a spatial coherence picture in the same period. The latest ERA5 and MERRA2, which include more recent instrument observations and improved data assimilation methods, should provide more reliable estimates than earlier studies. PSD (Power Spectrum Densities) analysis is used to analyze the time-scale of temporal variability and a significant test is used to diagnose the SAO signal is significant and not an artifact

e.g., just a harmonic part of the seasonal cycle. The energy budget, i.e., the heating rates according to dynamical, radiative, and moist processes, are analyzed to explain the formation of the SAO in temperature. To understand the relationship between SAO and SSTs, three model simulations with NCAR's Whole Atmosphere Community Climate Model, version 6 (WACCM6) are used. Details of the data and model simulations are described in section 2. In section 3, we show the analysis and results. Conclusions are summarized in section 4.



## 2 Data and Methods

### 2.1 GNSS RO temperature data

CHAMP provides ca. 150 occultation events globally per day from May 2001 to October 2008 and COSMIC began providing 1000-3000 occultation events per day since late 2006 (Wickert et al., 2001; Anthes et al., 2008). Many studies have demonstrated that GNSS RO temperature data have good quality in the range 8-30 km (Schmidt et al., 2005, 2010; Ho et al., 2009, 2012). In our study, we make use of monthly mean temperature data at 500-10 hPa. About 100 observations per month per 5° latitude grid can be provided by a single satellite CHAMP and more than 10 times the number of profiles are available since late 2006 due to the start of the COSMIC mission. WetPrf products are interpolated onto 100 m vertical resolution from 0.1 to 40 km based on the one-dimensional variational method (Kursinski et al., 2000; Wee and Kuo, 2015). We use the reprocessed and post-processed RO data, which are stable and accurate observations for climate studies. The CHAMP wetPf2 version is 2016.2430 and the COSMIC-1 wetPrf products are 2013.3520 and 2016.1120.

GNSS RO monthly zonal means of standard pressure level (400-10 hPa) were determined based on CHAMP and COSMIC-1 for the period 2001-2017, whereas 5° nonoverlapping latitude bands centered at 57.5°S–57.5°N were used. Data exceeding 3 times the standard deviation have been discarded at each level. For the COSMIC-1 data, we make additional monthly zonal means of standard pressure levels with a grid resolution 10° (latitude)x10° (longitude) for the period 2007-2017. The same averaging strategy is used.

Due to the missing data in the early part of the GNSS RO record, a spectral analysis of temperature data was performed using a lomb-scargle periodogram to identify the various periods in the time series (Lomb, 1976; Scargle, 1982; Horne and Baliunas, 1986; Press and Rybicki, 1989). For the SAO we choose the maximum power spectral density (PSD) between 5 and 7 months and use the probability 0.95 of detection for the significant test (Horne and Baliunas, 1986).

### 2.2 Reanalysis data

ERA5 is the latest ECMWF reanalysis (released in 2018) and various newly reprocessed data sets, recent instruments, improved data assimilation system are used in ERA5. Details information can be found in ERA5 data documentation https://confluence.ecmwf.int/display/CKB/ERA5%3A+data+documentation. The ERA5 monthly averaged data on pressure levels from 2001 to 2017 with 2.5° horizontal resolution is downloaded from Climate Data Store (CDS)(Hersbach et al., 2019). MERRA2 is the latest atmospheric reanalysis of NASA's Global Modeling and Assimilation Office (GMAO) with data resolution 0.5°x0.625° (Gelaro et al., 2017). Both reanalysis contain the eastward wind (u), northward wind (v), vertical pressure velocity ($\omega$) and temperature (T) on pressure levels. Both reanalyses have assimilated GNSS RO bending angles.

### 2.3 Model simulations

WACCM6 is one of the two available atmospheric components of the Community Earth System Model (CESM) from NCAR. It simulates atmospheric processes from the surface to about 140 km, which resolves the stratospheric dynamical and chemical



processes well (Marsh et al., 2013; Gettelman et al., 2019). The standard version of WACCM has 70 vertical levels with a vertical resolution of about 1 km in the UTLS region (Gettelman et al., 2019; Wang et al., 2019). The horizontal resolution used here is 1.9°x2.5°.

The model is integrated into its atmosphere-only mode, with prescribed SSTs. We first employ a Control simulation, with

SSTs prescribed to historically observed values over the period from 2001 to 2017. The SSTs are provided by Hadley Centre SST data set (Rayner et al., 2003). For comparison, we then employ two sensitivity simulations, which removed the SAO of SSTs (SST-SAO) globally (rmSAO run) and in the Tropics (rmSAO-TP run). The SST-SAO is removed by a band-pass filter (signals between 5 and 7 months are removed) using the Butterworth method (Butterworth, 1930). The differences between these two sensitivity simulations and the Control simulation, therefore, indicate the influences of the SST-SAO on the UTLS-

SAO.

## 2.4 Thermal budget analysis

To explain the SAO of temperature in the UTLS, the thermal budget is analyzed using the MERRA2 reanalysis as well as model simulations. The thermal budget of the UTLS is a balance between the dynamical heating and the total diabatic (Gettelman and Birner, 2007). The thermodynamic balance formalism is expressed as (Andrews et al., 1987; Abalos et al., 2013):

$$\bar{T}_t = -\bar{v}^*\bar{T}_y - \bar{\omega}^*S - e^{\frac{z}{H}}[e^{\frac{-z}{H}}(\overline{v't'}\frac{\bar{T}_y}{S} + \overline{\omega'T'})]_z + \bar{Q} \tag{1}$$

$$\bar{v}^* = \bar{v} - \rho_0^{-1}[\rho_0\overline{v'\Theta'}/\bar{\Theta}_z]_z \tag{2}$$

$$\bar{w}^* = \bar{w} + (a\cos\phi)^{-1}[\cos\phi\overline{v'\Theta'}/\bar{\Theta}_z]_\phi \tag{3}$$

where H is 7 km, $\rho_0$ is the atmosphere density, $\phi$ is latitude, overbars in the equation denote zonal means, primes indicate deviations from it and subscripts denote partial derivatives. $\bar{v}^*$ and $\bar{\omega}^*$ are components of the residual circulations, which can

be calculated by eq.2-3. $\Theta$ denotes potential temperature. $S = H\frac{N^2}{R}$ with R is $287m^2s^{-2}K^{-1}$ and $N^2$ is the Brunt-Väisälä frequency. The first three terms on the right side of the eq.1 indicate the dynamical heating, while Q is the diabatic heating. The dynamical heating includes the heating by the meridional component of the residual circulation (the first term on the right side of eq.1), the heating by downwelling (the second term on the right side of eq.1), and heating related to eddys (the third term on the right side of eq.1, mainly associated with the vertical eddy heat flux). In the UTLS region, Q is mainly determined

by radiation and moisture condensation processes.

MERRA2 provides very detailed temperature tendencies related to dynamics, radiation, moist, friction, gravity wave drag and near-surface turbulence. In the UTLS region, the friction, gravity wave drag and near-surface turbulence terms are very small, then we only use the dynamics, radiation and moist terms in this study. In addition, to keep a balance of the thermal budget during the data assimilation, an extra term called the analysis tendencies (ANA) is used in MERRA2 reanalysis (Mapes

and Bacmeister, 2012). ANA can be interpreted as the negative of model physical tendency error. Therefore, the ANA can be considered as residual (Mapes and Bacmeister, 2012). We further calculate the first two terms in eq.1 with the given equations and then separate the dynamical term of MEERA2 into the meridional component of the residual circulation, the heating by downwelling and eddy terms.



In the WACCM6 model, the temperature tendencies related to dynamics, radiation, moist and gravity wave drag processes
are also diagnosed and have direct outputs (Gettelman and Birner, 2007). Here we also use the dynamics, radiation and moist
terms in this study. The radiation term consists of long-wave and short-wave terms in the simulation. The dynamical term is
also separated into the meridional component of the residual circulation, the heating by downwelling and eddy terms with the
same method as MERRA2.

## 3   Analysis and Results

### 3.1   Spatial distribution of the SAO

According to the Lomb-Scargle PSD values for the temperature data, the annual cycle is strongest in most of the regions.
Figure 1 shows the SAO PSD in the period 2001-2017 and the lapse rate tropopause determined from GNSS RO data. The
most prominent SAO signal with strong and significant SAO PSD can be seen in the UTLS region (250-175 hPa) from the
subtropics (22.5°) to mid-latitudes (42.5°) in both hemispheres. This is also observed in the ratio of semiannual/annual cycle
PSD (Figure S1). The ratio is larger than 0.6 in the UTLS (300-175 hPa) from the subtropics to mid-latitudes (20-45°) and
middle-upper troposphere (500-200 hPa) from tropics (10°S-10°N). The most prominent SAO PSDs can also be seen in the
tropics from 400 to 225 hPa (Figure 1 and Figure S1). This is consistent with the results from Loon and Jenne (1969), which
shows clear SAO of temperature in the upper troposphere in the tropics and SH subtropics. It is also consistent with the results
from Fueglistaler et al. (2009), which shows that there is no significant annual cycle in the tropical upper troposphere. Our
results give a survey of the SAO and find that the significant SAO also persists in the lower stratosphere and the mid-latitudes,
i.e., 70-30 hPa in SH (32.5°S-47.5°S) and NH (37.5°N-47.5°N) mid-latitudes. It is also noteworthy that the magnitude of SAO
PSD in the tropics in the troposphere and lower stratosphere are much weaker than the signals in mid-latitudes. We then mainly
focus on the SAO in the UTLS region (250-175 hPa) in mid-latitudes (22.5°-42.5°) hereafter in this study.

The reanalysis data and model simulations show good agreement with GNSS RO for the general pattern of SAO PSD. The
ERA5 data show the best agreement with the GNSS RO, with a very similar spatial pattern and comparable magnitude as
shown in Figure 1a-b. The SAO signals in MERRA2 are slightly weaker and less significant than those in GNSS RO and ERA5
(Figure 1c and Figure S1c). The model also shows a good representation of the SAO in the UTLS, except that the significant
region of SAO in the tropical regions is not observed from 350 to 200 hPa (Figure 1d and Figure S1d).

Further information can be gained from Figure 2, which shows the latitude-longitude distributions of the SAO PSD at 200
hPa. The significant SAO region at 200 hPa in mid-latitudes is larger in the Eastern than in the Western Hemisphere, which
is consistent with previous studies (Loon and Jenne, 1969). Note that due to data limitation, the PSD shown in Figure 2a
is calculated for the period 2007-2017. Further analysis indicates that the different period of analysis does not influence the
results. As shown in Figure S2, the PSD distribution 2007-2017 from the ERA5, MERRA2 and the Control simulation is
very similar to that shown in Figure 2, which is analyzed for the period 2001-2017. Large SAO PSD occurs around the Asian
monsoon region, the Australian monsoon region, as well as the North American monsoon region. The most prominent SAO
signal overlies on the three monsoon regions indicate that the UTLS SAO is related to the monsoon processes (Chen and Wu,





1992). A further energy budget analysis indicates that the vertical transportation of water vapor and subsequent condensation in the upper troposphere over the monsoon regions (e.g., the Asia monsoon region) is much stronger than other regions and leads to a stronger peak of temperature in the summer season (Figure S3). At the same time, the radiative cooling is relatively

weaker than other regions, which also contributes to the stronger temperature peak in the Asia monsoon region. Again, the ERA5 has the best agreement with GNSS RO measurements with a consistent pattern and comparable magnitude (Figure 2b). The MERRA2 and the model simulation also show good agreement with the GNSS RO data (Figure 2c-d). It is also noteworthy that there are also significant SAO signals in the tropics except over the Pacific. However, the magnitude of the SAO PSD in the tropics is much weaker than that in mid-latitudes. According to previous studies, the SAO of the tropical stationary eddies

is caused by the semi-annual east-west seesaw of the global divergent circulation between the areas of the Asian-Australian (AA) monsoon (60°E-120°W) and the extra-AA monsoon (120°W-60°E) particularly in SH tropics (Chen and Wu, 1992; Chen et al., 1996).

### 3.2    Time evolution and mechanism of the SAO in the UTLS

To analyze the time evolution of SAO signals, Figure 3 shows the time series in the zonal mean temperature and corresponding

annual cycle anomalies in the NH mid-latitudes (32.5°N-42.5°N) and SH Mid-latitudes (32.5°S-42.5°S) at 200 hPa. From the time series, the SAO of temperature is evident in both hemispheres, with two clear peaks of temperature in each year. Such oscillation can be seen in all the observed and reanalysis data sets, including the GNSS RO measurements as well as the ERA5 and MERRA2 reanalysis. The model simulations can also simulate the SAO of temperature except that an offset is of about 1 K between the Control simulation and the observation/reanalysis. From the annual cycle anomalies, two peaks of temperature

exist in February and July in the NH and in February and August in the SH. The peak in July/August is more intense than in February, especially in the NH. For all the datasets, the peaks in January-February and July-August are quite even in NH. In SH the peak at August of simulations is weaker than observation/reanalysis.

To further investigate the mechanism of the SAO in the UTLS, the thermal budget is analyzed using the MERRA2 data and model simulations. The annual cycle of the heating rates at 200 hPa averaged over the latitude bands from 32.5° to 42.5° in

two hemispheres is shown in Figure 4. To show the relationship between the heating rates and the temperature, the annual cycle of temperature from the MERRA2 reanalysis data is also shown in Figure 4. In the NH, there are two peaks of total heating rates in December and June and two nadirs in March and September (Figure 4a). While the total heating rate is positive, the temperature will increase. The total heating rates are positive from mid-November to mid-January and from April to July, which leads to increases in temperature during the same periods. From mid-January and July, the total heating rate turns to be

negative, and the temperature starts to decrease. Therefore, the are two peaks of temperature in February and July. At the same time, there are two nadirs of temperature in April and November. To clearly show the relative contribution of different terms to the total heating rate, the variation of each term at peaks and nadirs of total heating rates is selected to show in Figure 5. Over all, the dynamical processes contribute the most prominent positive heating rates through all terms, which is mostly offset by the radiative cooling. The radiative term can further be separated into short-wave warming and long-wave cooling (Figure

S4). The short-wave and long-wave terms are all strongest during summer and the radiation heating rates are negative due



to larger negative value of long-wave terms. The moist and analysis terms are quite small. For the first temperature peak in Jannuary-February, it is mainly related to the dynamical heating, since the dynamical warming is slightly stronger than the radiative cooling and other terms are near zero in December. The second temperature peak in July is caused by a combination of moist and dynamical processes. There is a secondary peak of dynamical heating in June, although it is offset by radiative cooling. The moist processes delay the transition of total heating rates from positive to negative, which is in July but would be a month earlier otherwise. This positive peak in moist heating is related to the release of heat by the condensation of water vapor that is transported by deep convections in the upper troposphere. The negative total heating rates in March and September are mainly determined by the strong radiative cooling, which is stronger than the sum of other terms.

In the SH (Figure 4b and Figure 5b), the total heating rates are positive from mid-November to mid-January and from May to August, which leads to a first peak of temperature in February and the second peak in August. For the SH summer, although the annual variation of the moist heating is relatively weak, it is stronger in DJF than in other seasons. In addition, the dynamical heating is also relatively strong in DJF. Therefore, both the dynamical and moist processes contribute to the positive total heating rates and lead to the temperature peak in SH summer. For the SH winter, the positive total heating rate and subsequent temperature maximum are mainly related to dynamical heating. Note that the ANA term in the SH is large, which indicates a poor ability of the MERRA2 model to represent the temperature variations in SH UTLS. The moist processes are lower in the SH than in the NH, which is perhaps due to overall weaker monsoon circulation. The energy budget analysis can also be confirmed by model simulations (Figure 6). Figure 6 is the same as Figure 4 but with Control simulation. The model shows a very similar annual variation of dynamical, radiative, moist and total heating rates in both the NH and SH compared to the MERRA2 reanalysis.

In Figure 7 the dynamical term is further divided into the eddy, heating by downwelling ($-\omega S$) and heating by the meridional component of the residual circulation ($-vT_y$) terms according to Eq.1. The eddy term is mainly associated with the vertical eddy heat flux. As seen from Figure 7, the $-vT_y$ term is close to zero, which is less important in the UTLS region than other terms as also indicated by previous studies (Abalos et al., 2013). The $-\omega S$ term is strongest during boreal winter (DJF) in both hemispheres, since the vertical component of the residual circulation (including the tropical upwelling and the extra-tropical downwelling) is most prominent in DJF. This is consistent with previous studies which shows a strongest dynamical cooling according to the upwelling in the tropics Abalos et al. (2013). In the SH, there is a secondary peak of the $-\omega S$ term in JJA. The eddy term is non-negligible in the upper troposphere (at 200 hPa) and peaks from April to June. The first peak in dynamical heating during boreal winter (DJF) is dominated by the $-\omega S$ term while the secondary peak near June is significantly influenced by the $-vT_y$ term.

## 3.3 Relationship between the UTLS-SAO and the SSTs

As introduced in the sect. 1, there is also a pronounced SAO signal at the surface (Meehl et al., 1998). While the surface is the main energy source of the atmosphere, it would be very interesting to investigate the relationship between the UTLS-SAO and the Surface-SAO. Since the surface-SAO is a coupled ocean-atmosphere phenomenon and is strongly modified by the seasonal cycle of SSTs (Meehl et al., 1998), we mainly focus on the relationship between the UTLS-SAO and the SAO in SSTs (SST-





SAO) in this study. We checked the PSD of the SST-SAO (Figure S5), and there are significant SAO signals in the tropics from the Easter Pacific (120°W) to the Atlantic Ocean, Indian Ocean and the Western Pacific (150°E), which is also found by earlier studies (Schott et al., 2009; Park and Lee, 2014; Yan et al., 2018). The SAO signals over the Central Pacific (150°E-120°W) are not significant (Figure S5), which resembles the UTLS-SAO in the tropics as shown in Figure 2. This indicates the potential connection between the SST-SAO and the UTLS-SAO in the tropics.

To identify the relationship between the subtropical to mid-latitude UTLS-SAO and the SST-SAO, the correlation coefficients are calculated between the tropical(5°S-5°N), SH/NH mid-latitude UTLS-SAO and the SST-SAO (Figure S6). Firstly, we extracted ERA5 temperature SAO signals (5-7 months) using the 1-D wavelet Morlet transform (Lilly and Olhede, 2009, 2012). Secondly, we calculated the Pearson correlation coefficient between SAO signals and SST-SAOs. As shown in Figure S6, both the SH/NH mid-latitude and tropical UTLS-SAO are strongly and significantly correlated with SST variations. The UTLS-

SAO shows a negative correlation with SST-SAO in the tropics (20°S-20°N), especially over the Indian Ocean and the Western Pacific, whereas the SST-SAO are most pronounced (Figure S6). The SH/NH mid-latitude and tropical UTLS-SAO show very similar correlation patterns with SSTs since the UTLS-SAO in the two hemispheres are almost in phase with each other (Figure 3).

We then further investigate the impacts of the SST-SAO on UTLS-SAO using our model simulations. As introduced in

Section 2.3, we first employ a Control simulation using the WACCM6 model with SSTs prescribed to observed variations. As seen in Figures 1-3, the model represents the UTLS-SAO pretty well although the simulated temperature is ca. 1 K lower than observations. We then employ a sensitive simulation with the same configuration except that the SST-SAO has been removed globally (the rmSAO run). As discussed above, SST-SAO is mainly significant in the tropics (Figure S5), and the UTLS-SAO is most correlated with tropical SSTs, therefore we employ a third simulation with the SST-SAO removed only in the tropics

(the rmSAO-TP simulation).

Figure 8 shows the PSD of SAO in temperature from the rmSAO and rmSAO-TP runs, as well as their relative differences with the Control simulation. Similar to the Control simulation, there are significant SAO signals in the rmSAO and rmSAO-TP simulations. However, while the SST-SAO is removed globally, the SAO in the tropical upper-middle troposphere (500-175 hPa) is not significant, which indicates that the upper-middle troposphere SAO in the tropics is significantly influenced by

the SST-SAO. If the SST-SAO is removed only in the tropics, the upper troposphere SAO in the tropics is still significant. This demonstrates that the SST-SAO in extra-tropics is important. At the same time, the SAO in the lower stratosphere of mid-latitudes (70-30 hPa in SH 32.5°S-47.5°S and NH 37.5°N-47.5°N) is also significantly reduced in both the rmSAO and rmSAO-TP simulations.

Although the UTLS-SAO is still significant in the rmSAO and rmSAO-TP simulations, the magnitudes of the UTLS-SAO

in the two sensitive simulations are significantly reduced (Figures 8c-d). Compared with Control simulation, the magnitude of rmSAO simulation is reduced by 31% for NH and 55% for SH (rectangular box in Figures 8c). The averaged magnitude reduction of rmSAO-TP simulation is 14% in NH and 41% in SH (rectangular box in Figures 8d). Such reduction of SAO PSD caused by removing SST-SAO is more evident in the SH and NH than tropics. Compares the magnitude of the reduced PSD in





Figures 8c-d and the PSD in three model simulations (Figure 1d, 7a-b), about 31% of the PSD can be reduced by removing the
SST-SAO in NH and about 55% in SH. This clearly shows that the SST-SAO modifies the strength of the SAO in the UTLS.

This result can also be confirmed by the averaged seasonal cycle of the extracted UTLS-SAO signal as shown in Figure 9.
To compare the amplitude of SAO signals in model simulations, we extracted SAO signals (5-7 months) using the 1-D wavelet
Morlet transform (Lilly and Olhede, 2009, 2012). As shown in Figure 9, the averaged amplitude of the rmSAO simulation
(blue lines) is smaller than the Control simulation (red lines) (11%), and the amplitude of the rmSAO-TP simulation (dash
black lines) is slightly weaker than the Control simulation (1%). This demonstrates that the SST-SAO in extra-tropics for the
UTLS-SAO in NH is more important. Reduction of the SAO amplitude related to removing SST-SAO is more evident in the
SH (Figure 9b). Compared with the Control simulation, the averaged amplitude of the rmSAO and rmSAO-TP simulation is
reduced by 33% and 29%, respectively. This might be related to the relatively large area of ocean in the SH compared to the
NH. Anyway, a connection has been established between the surface and the UTLS SAO, and it is obvious that the SST-SAO
influences the UTLS-SAO significantly.

To further find out through what processes of the SSTs modifies the UTLS-SAO, the thermal budget analysis is applied
to three model simulations. Figure 10 shows a comparison of the thermal budget in the UTLS between the two sensitivity
simulations and the Control simulation. In the NH, the total heating rates are significantly reduced from May to July, which
is mainly caused by the weaker dynamical and moist heating in the two sensitivity simulations. From August to September,
the reduction of moist heating is also evident but is offset by less radiative cooling. In October, the negative values of the total
heating rates are also reduced in the two sensitivity simulations due to the reduced radiative cooling. During boreal winter, the
impact of removing SST-SAO is not significant.

In the SH, the total heating rates are reduced overall and especially significantly reduced in summer (November-December).
This is related to the weaker dynamical and moist heating, although it is partly offset by less radiative cooling. In winter, the
changes in total heating rate are not significant either. In April, the negative values of the total heating rate are also reduced due
to the reduced radiative cooling. In summary, the UTLS-SAO is modified by the SST-SAO mainly through its modification to
the summer time moist heating and autumn radiative cooling.

## 4   Conclusions

Variability of zonal mean temperatures over 500-10 hPa is analyzed based on high-quality GNSS temperature measurements
and reanalysis data (MERRA2 and ERA5) covering 2001-2017. All data sets show a good agreement and ERA5 has the
best agreement with GNSS RO data. The model simulations have also good agreement with other data sets except for a
relatively large difference with other data sets in the tropical region. Our results confirm the tropical and subtropical SAO
of temperature in the upper troposphere as reported by previous studies (Loon and Jenne, 1969). In addition, we also find a
significant SAO of temperature in the lower stratosphere (just above the tropopause and from 70 to 30 hPa) in the mid-latitudes
of both hemispheres. From the PSD of the SAO of temperature, the SAO signal is most pronounced from the subtropics to mid-
latitudes ( 22.5°-42.5°) in the UTLS region (250-175 hPa). The SAO of temperature with maxima in February and July/August





in the NH/SH mid-latitudes in the UTLS is observed. Furthermore, the most prominent SAO signal overlies on the monsoon regions indicate that the UTLS SAO is related to the monsoon processes.

The explanation of the SAO of temperature in the UTLS is given by a thermal budget analysis. In the NH, the first temperature

peak is in February is mainly related to the dynamical heating while the second peak in July is caused by a combination of moist and dynamical processes. The temperature peaks follow the transitions of total heating rates by definition, and the moist processes delay the transition of total heating rate from positive to negative in July. In the SH, both the dynamical and moist processes contribute to the temperature peak in summer, while the temperature maximum in winter (August) is mainly related to dynamical heating. The further energy budget analysis indicates that the vertical transportation of water vapor and subsequent

condensation in the upper troposphere over monsoon regions is much stronger and leads to a stronger peak of temperature in the summer season. Through the thermal budget analysis, we also find that the UTLS-SAO in the SH can be well represented by the WACCM6 model. A significant part in the SH represented by MERRA2 model cannot be explained by ANA, dynamical and moist process, which indicates that the budget for the SH is not as reliable as the NH.

Based on a series of model simulations, we analyze the relationship between UTLS-SAO and the SST-SAO. The results

indicate that the amplitude of the UTLS-SAO can be significantly reduced while the SST-SAO is removed in the tropics, and will be further reduced if the SST-SAO is removed globally. The UTLS-SAO of temperature is therefore partly modified by the SSTs. A comparison of the thermal budget between the sensitivity simulations and the Control simulation further indicates that the SST-SAO mainly modifies the UTLS-SAO through its modification to the summer time moist heating and autumn radiative cooling.

The annual and semi-annual variations are the most pronounced short-term climate signals. In the past decades, numerous attentions have been attracted to investigate various aspects of the semi-annual cycle in the surface and stratosphere. Few attempts have been made to analyze the SAO in the UTLS especially in the mid-latitudes. The major effort of this study not only explores the structure of the SAO in the mid-latitudinal UTLS and its relationship with thermodynamic balance but also provides the relevance between the SAO in the UTLS and SSTs by climate model simulations. However, a full mechanistic

understanding of the SAO drivers in the UTLS, the predictability of the SAO in the UTLS as well as its potential impacts on surface weather and climate awaits further studies.

*Data availability.* The authors are greatly appreciative to the teams of the CDAAC for the use of the GNSS RO data sets (http://cdaac-www.cosmic.ucar.edu/cdaac/index.html), the NASA GSFC for MERRA2 data (https://disc.gsfc.nasa.gov/datasets?project=MERRA-2) and the Copernicus Climate Change Service and the ECWMF for the ERA5 data (https://cds.climate.copernicus.eu). The simulations can be

provided to readers by contacting the corresponding author.



*Author contributions.* MS performed the computational implementation and the analysis, created the figures and wrote the first draft of the paper. WW made the model simulations, created the figures and provided advice on the analysis design and contributed to the text. All authors contributed to the study design.

*Competing interests.* The authors declare that they have no conflict of interest.

*Acknowledgements.* The research has been supported jointly by the National Natural Science Foundation of China (grant no. 41904023 and 42075055), and the Fundamental Research Funds for the Central Universities, China University of Geosciences (Wuhan) CUG2106357.



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

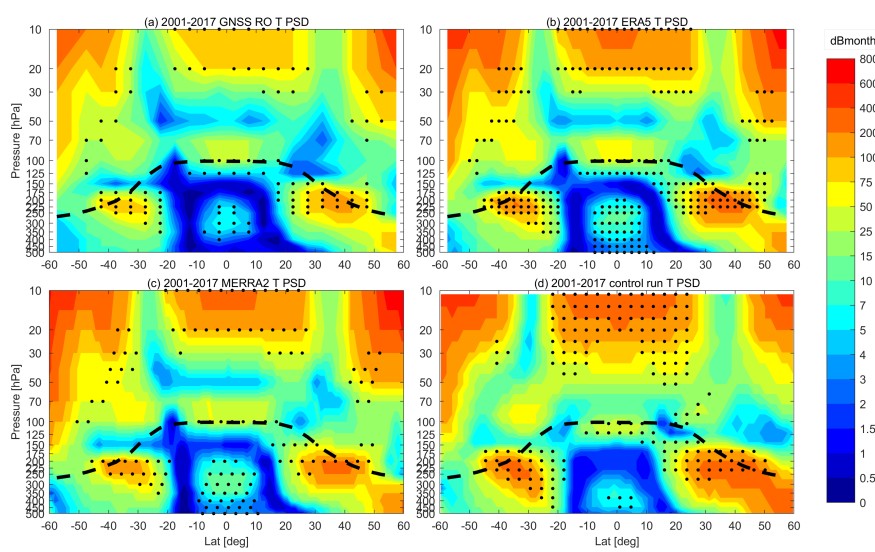

**Figure 1.** The power spectrum densities (PSD) of SAO based on GNSS RO (a), ERA5 (b), MERRA2 (c) and model simulation (d) temperature for the period 2001-2017. The dots mark the significant area at 95% level. The dashed black lines mark the tropopause height calculated with GNSS RO data.



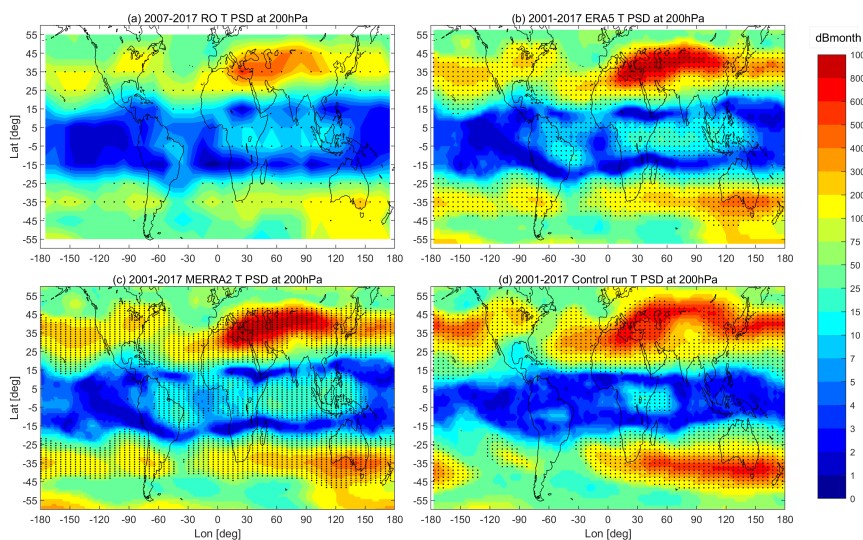

**Figure 2.** The PSD of SAO based on COSMIC-1 (2007-2017) (a), ERA5 (2001-2017) (b), MERRA2 (2001-2017) (c) and model simulation (2001-2017) (d) at 200 hPa. The dots mark the significant area at 95% level.

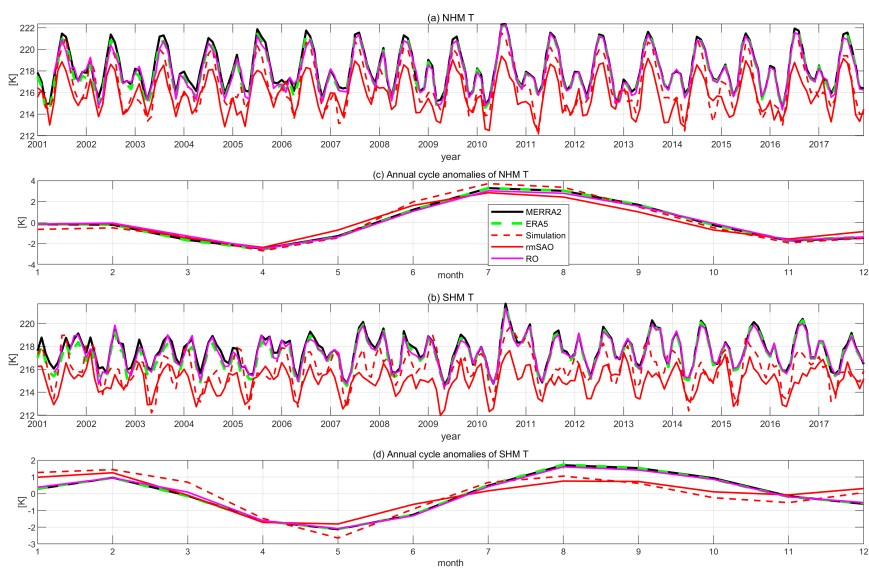

**Figure 3.** Time series of the zonal mean temperature (T) at 200 hPa averaged around the Northern Hemisphere mid-latitudes (NHM) 32.5°N-42.5°N (a) and their corresponding annual cycle anomalies (b). The Southern Hemisphere mid-latitudes (SHM) 32.5°S-42.5°S (c) and their corresponding annual cycle anomalies (d).



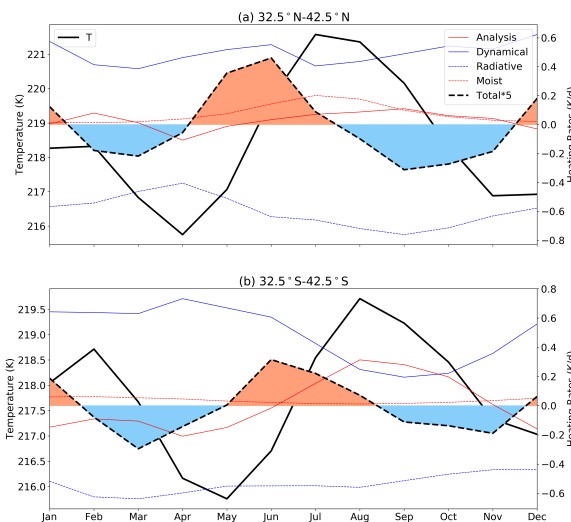

**Figure 4.** Annual cycle of the zonal mean temperature (T) and heating rates at 200 hPa averaged around the NHM 32.5°N-42.5°N (a) and the SHM 32.5°S-42.5°S (b) based on MERRA2 data. The red, blue, dashed blue and dashed red lines indicate the heating rates related to dynamics, radiation, condensation and analysis processes, respectively. The positive total heating rates are filled with light red color and the negative total heating rates are filled with light blue color. The total heating rates, which are the sum of analysis, dynamical, radiative and moist heating rates, have been 5 times enlarged to be more visible in the figures.





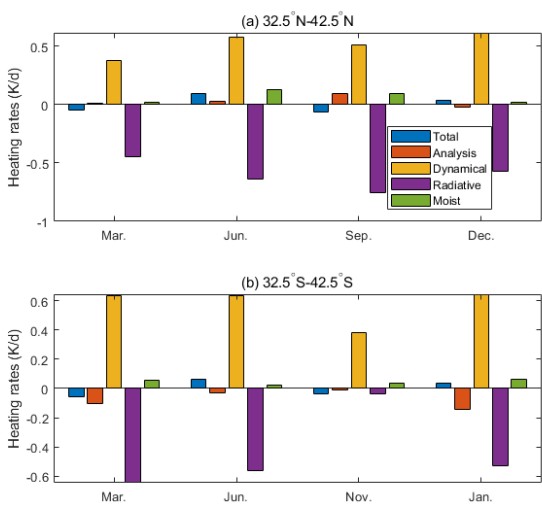

**Figure 5.** The variation of heating rates terms at 200 hPa at peaks and troughs of the total heating rates around the NHM 32.5°N-42.5°N (a) and the SHM 32.5°S-42.5°S (b) based on MERRA2 data. The blue, red, yellow, purple and green columns are total, analysis, dynamical, radiative and moist terms, respectively.



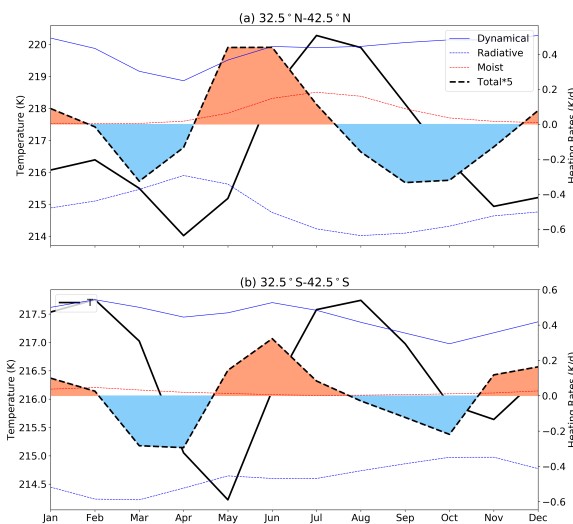

**Figure 6.** Annual cycle of the zonal mean temperature (T) and heating rates at 200 hPa averaged around the NHM 32.5°N-42.5°N (a) and the SHM 32.5°S-42.5°S (b) based on Control simulation. The blue, dashed blue and dashed red lines indicate the heating rates related to dynamics, radiation and condensation processes, respectively. The positive total heating rates are filled with light red color and the negative total heating rates are filled with light blue color. The total heating rates, which are the sum of dynamical, radiative and moist heating rates, have been 5 times enlarged to be more visible in the figures.



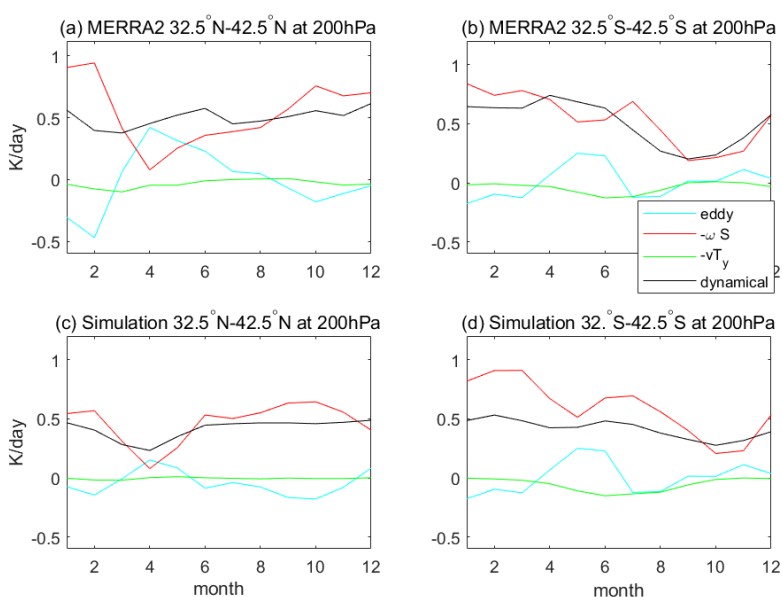

**Figure 7.** Annual cycle of dynamical heating rates at 200 averaged around the NHM 32.5°N-42.5°N (a) and the SHM 32.5°S-42.5°S based on MERRA2 (a-b) and Control simulation (c-d). The cyan, red, green and black lines are the eddy , heating by downwelling (-$\omega$S), heating by the meridional component of the residual circulation (-$vT_y$) and dynamical term.



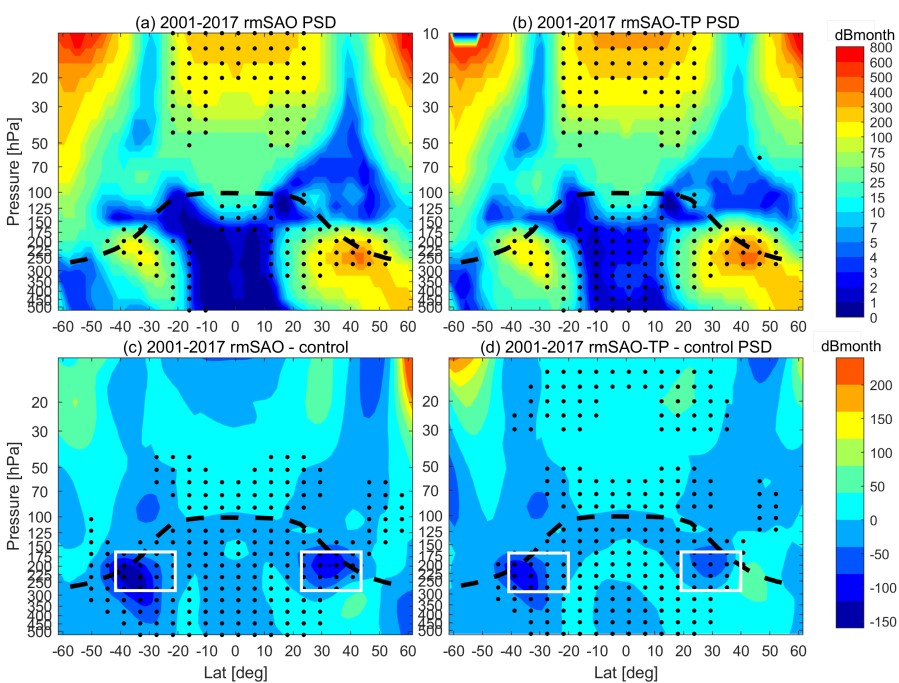

**Figure 8.** (a) The PSD of SAO analyzed for the period 2001-2017 based on model simulation with removed SST-SAO (rmSAO). (b) Same as (a), but for the simulation with removed SST-SAO in the tropics (rmSAO-TP). The dots mark the significant area at 95% level. (c) The relative difference of SAO PSD between the rmSAO and the Control simulations (rmSAO - Control) for the period 2001-2017. (d) Same as (c), but for the difference between the rmSAO-TP and the Control simulations (rmSAO-TP - Control). The black dots mark area with significant difference of the two time series at 95% level based on t-test. The dashed black lines mark the tropopause height calculated with GNSS RO data.

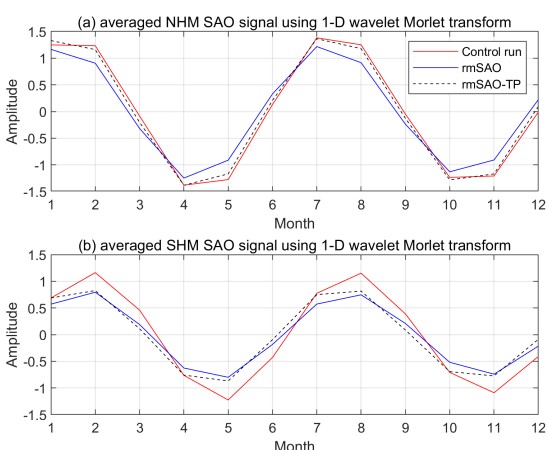

**Figure 9.** The averaged of the extracted SAO singals (5-7 months) based on 1-D wavelet Morlet transform for the Northern Hemisphere (NH) mid-latitudes (a) and for the Southern Hemisphere (SH) mid-latitudes. The blue, red and dashed black lines are results of the Control, rmSAO and rmSAO-TP runs, respectively.



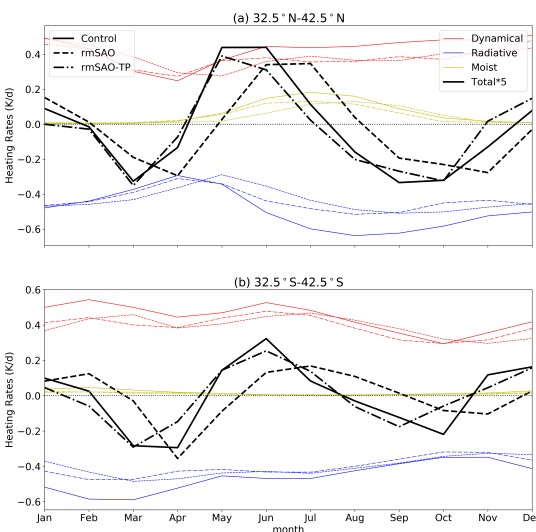

**Figure 10.** Annual cycle of the heating rates at 200 hPa averaged around the Northern Hemisphere mid-latitudes 32.5°N-42.5°N (a) and the Southern Hemisphere mid-latitudes 32.5°S-42.5°S (b). The red, blue and yellow lines indicate the heating rates related to dynamics, radiation and condensation processes, respectively. The total heating rates, which are the sum of the dynamical, radiative and moist heating rates, are illustrated by black lines. The solid, dashed and dotted-dashed lines indicate data from the Control, rmSAO and rmSAO-TP simulations, respectively.