# Peer review of "Figure S1: The ratio between SAO and annual cycle PSD based on COSMIC-1 (2007-2017) (a), ERA5 (2007-2017) (b), MERRA2 (2007-2017) (c) and model simulation (2007-2017) (d). The dots mark the significant area at 95% level. The dashed black lines mark the tropopause height calculated with GNSS RO data."

_Atmospheric Chemistry and Physics, 2022_

## Referee Comment (RC2)

Review of the paper:

"The semi-annual oscillation (SAO) in the upper troposphere and lower stratosphere (UTLS)"

written by Shangguan and Wang,

**General comments:**

This paper examines the semi-annual oscillation (SAO) of temperature in the upper troposphere and lower stratosphere (UTLS) using the data from satellite observations, two reanalyses, and model simulations. Furthermore, this paper investigates the thermodynamic components which affect the SAO and find the connection between SAO in the surface and SAO in the UTLS. Overall, this is an interesting and well written paper. The manuscript contains material that is of interest to ACP readers. However, there are a few issues that I would like to see addressed before it is suitable for publication.

**Specific comments:**

- Figure 1, Figure 2, and Figure 8
  It is really difficult to understand Figure 1, Figure 2, and Figure 8. I was confused the first time that I read the paragraphs regarding these three figures. The color shading in the figures seems to indicate some variable related to the time period from the unit 'dBmonth' in the figures. However, the texts related to these figures describe the color shading as the strength or magnitude of temperature SAO. The PSD method is briefly mentioned in the end of Section 1. Since several figures and a lot of the results in the manuscript are based on the PSD method, it might be worth introducing the PSD method in more detail and explaining the meaning of 'dBmonth' to guide the readers. If possible, it might be better to connect 'dBmonth' with 'Kelvin' unit.

- L138-L139
  I'm not sure how the result is consistent with the results from Fueglistaler et al. (2009)?

- L154-L160
  The UTLS SAO is significant almost in the whole zonal band in the middle latitude. Monsoon might amplify the amplitude of the SAO inside of some monsoon regions, but it does not suggest the direct connection between monsoon and the UTLS SAO outside of monsoon regions from Figure 2. Especially, the authors use Asian summer monsoon as an example to prove the influence of the transportation of water vapor and its condensation on th SAO. Does this also apply in the connection between Australian region and SAO? Maybe the authors should clarify the statement.

- The authors mentioned "We then mainly focus on the SAO in the UTLS region (250-175 hPa) in mid-latitudes (22.5 °-42.5°) hereafter in this study." in L142-143. However, almost all the results are about the region in 32.5 °-42.5° since Section 3.2. An explanation is required for taking different regions.

- L207-L209 and Figure 6

  There is no new information related to Figure 6 described in the manuscript. I would suggest to remove Figure 6 here and put it in the supplement.

- L219

  "by the -vTy term", should it be "by the eddy term"? Please check

- L236-L238

  The nearly concurrent UTLS-SAO in the two hemisphere can explain the similar correlation patterns between SH and NH mid-latitude with SSTs, but not between SH/NH mid-latitude and tropical UTLS-SAO with SSTs.

- L248-L251

  "However, while the...in extra-tropics is important.", I would suggest to change this sentence as "While the SST-SAO is removed globally, the SAO in the tropical upper-middle troposphere (500-175 hPa) is not significant. However, if the SST-SAO is removed only in the tropics, the upper troposphere SAO in the tropics is still significant, which indicates that the upper-middle troposphere SAO in the tropics is significantly influenced by the SST-SAO in the extra-tropics".

- L257-L258

  "Such reduction of..." Maybe the reduction of the absolute magnitude is more evident in the SH/NH compared to that in the tropics. The relative reduction seems comparable given the weak SAO in tropics in the control simulation.

- L258-L260

  This sentence just repeats the conclusion in L255-L256. Additionally, it should be "Figure 1d, 8a-b"

- L281-L282 and L308

  The modification in the summer moist heating is really small compared to that in the dynamical heating and radiative cooling for SH. I am not sure it is appropriate to make this conclusion for SH.

**Technical comments:**

- L8

  "the winter time ... the summer time maximum"–> "the austral winter time ... the austral summer time maximum"
  To avoid the confusion, please check and add "austral" in front of winter and summer for SH in the whole manuscript

- L82

  Details information –> Detailed information

- L185

  Therefore, the are –> Therefore, there are

- L187
  in Figure 5 –> Figure 5a

- L190
  Figure S4 –> Figure S4a

- L216
  Abalos et al. (2013) –> (Abalos et al., 2013)

- L231
  "between the tropical(5°S-5°N), SH/NH mid-latitude UTLS-SAO and the SST-SAO" might be better to change as "between the SST-SAO and the UTLS-SAO over the tropical(5°S-5°N) and the SH/NH mid-latitude"

- L236
  Pacific, whereas the SST-SAO are most pronounced (Figure S6) –> Pacific (Figure S6a), whereas the SST-SAO are most pronounced (Figure S5a)

- L256
  31% for NH and 55% for SH –> 31% for NHM and 55% for SHM
  Please check and replace NH/SH with NHM/SHM through the whole manuscript when it is necessary, and add abbreviation about NHM/SHM in proper place ahead

- L263
  in Figure 9 –> in Figure 9a

- L267
  SH (Figure 9b) –> SH (Figure 9b) compared to that in the NHM

- L286-L287
  except for a relatively large difference with other data sets in the tropical region –> except in the tropical region

---

## Author Response (AR1)

**Response to Anonymous RC #1 's Comments,**

The paper by Shangguan and Wang is an interesting piece of work that addresses the semiannual oscillation (SAO) in the UTLS. This topic is of relevance because of its potential to provide predictive capabilities on seasonal and subseasonal time scales. Another merit of this work is that the SAO in the UTLS region has not been topic of many studies so far. The authors investigate the UTLS SAO in temperatures of two reanalyses, temperatures observed by GNSS RO, and simulations with WACCM. In a sensitivity simulation, the influence of the SST SAO on the UTLS SAO is examined.

The topic of this paper falls into the scope of ACP and is of interest to its broad readership. Generally, the paper is well written. The paper is therefore recommended for publication in ACP after addressing my mainly minor comments that are given below.

Response:We would like to sincerely thank the referee for the constructive comments and suggestions. Following the reviewer's suggestion, we have substantially revised the manuscript. Please note that the manuscript is also altered according to the other reviewer's comments and suggestions. More details can be found in the point-by-point responses as shown below.

Main Concern:

The authors claim that regions where the UTLS SAO is strongest would coincide with the global monsoon regions, and it would therefore indicate that the UTLS SAO is related to the monsoon processes. However, this is not supported by the figures shown in the paper. The UTLS SAO has even minima in the monsoon regions. It maximizes roughly at 35N and 35S, and in some regions near the equator.

Response: Thank you very much for the very helpful comment. We agree to the reviewer that the UTLS SAO does not coincide with the monsoon regions. Based on further analysis, we think that the strong SAO signals over that region are likely related to both moist and dynamical processes. More detailed discussions can be seen in our response to your specific comments (10). The relevant content has been modified after careful consideration.

Specific comments:

(1) I.22: This is not entirely correct: The MA-SAO is a complex interplay of momentum advection, planetary waves from the extratropics, and vertically propagating waves (both global-scale waves and small-scale gravity waves) (Hamilton and Mahlmann, 1988; Richter et al. (2006); Ern et al. (2015, 2021)).

Hamilton, K. and Mahlmann, J. D.:General Circulation Model simulation of the semiannual oscillation of the tropical middle atmosphere, J. Atmos. Sci., 45, 3212-3235, 1988.

Ern, M., Preusse, P., and Riese, M.: Driving of the SAO by gravity waves as

observed from satellite, Ann. Geophys., 33, 483-504, https://doi.org/10.5194/angeo-33-483-2015, 2015.

Ern, M., Diallo, M., Preusse, P., Mlynczak, M. G., Schwartz, M. J., Wu, Q., and Riese, M.:The semiannual oscillation (SAO) in the tropical middle atmosphere and its gravity wave driving in reanalyses and satellite observations, Atmos. Chem. Phys., 21, 13763-13795, https://doi.org/10.5194/acp-21-13763-2021, 2021.

Response:Thank you very much. We have corrected the sentence in the introduction according to your suggestion.

(2) l.23-36: There is some more work that should be mentioned in the introduction: In a recent paper, Yang and Wu (2022) relate the semiannual surface air response to changes in the oceanic mixed layers.

Yang, F., and Wu, Z.: On the physical origin of the semiannual component of surface air temperature over oceans, Climate Dynamics, https://doi.org/10.1007/s00382-022-06199-z, 2022.

Bracegirdle (2011) showed that at latitudes between 50S and 65S the SAO in sea level pressure is coupled with the stratospheric circulation, and correlations are found over a large altitude range in the troposphere and stratosphere. In particular, a downward influence from the stratosphere is significant in late summer/early autumn.

Bracegirdle, T. J.: The seasonal cycle of stratosphere-troposphere coupling at southern high latitudes associated with the semi-annual oscillation in sea-level pressure, Climate Dynamics, 37, 2323-2333, doi:10.1007/s00382-011-1014-4, 2011.

Response:Thank you very much. We have added suggested references in the introduction.

(3) l.55: Also harmonics of the annual cycle could result in a significant contribution to the semiannual signal. For example, if the annual variation is not a sinus-like variation, but instead consists of only a narrow peak, a whole series of harmonics would be needed to describe its temporal variation. Therefore you should emphasize that the SAO variations that you are investigating are more sinus-like, as you show in Fig.3.

Response:Thank you for your suggestions. We have modified the sentence.

(4) l.67: GNSS-RO temperatures of the wetPrf product may not be fully reliable anymore in the troposphere. For the wetPrf retrieval usually ECMWF analysis temperatures in lower resolution are used as input, and then a 1DVar retrieval is performed to disentangle the effect of humidity and temperature on RO bending angles in the troposphere (Kursinski et al., 2000). Therefore it is not completely clear to which degree the wetPrf product and ECMWF analyses are independent (perhaps this is the reason why GNSS RO and ERA5 agree so well). I guess, you are using GNSS-RO temperatures at altitudes above ~5-8km where they should

still be reliable. But still you should add a cautionary note for altitudes below ~5-8km, where biases could exist (see Xu et al., 2017).

References:

Kursinski, E. R., Healy, S. B., and Romans, L. J.: Initial results of combining GPS occultations with ECMWF global analyses within a 1DVar framework, Earth Planets Space, 52, 885-892, 2000.

Xu, G., Yue, X., Zhang, W., and Wan, X.: Assessment of Atmospheric Wet Profiles Obtained from COSMIC Radio Occultation Observations over China, Atmosphere, 8, 208, doi:10.3390/atmos8110208, 2017.

Response:Yes, we use wetPrf data between 500 and 10 hPa (above 5 km) and we have added the reference and the cautionary note in the manuscript.

(5) l.110: Please note that w* is not the pressure vertical velocity!
Please mention the conversion between w* and omega* that you are using!

Response:Thank you very much for the reminder. We have changed the corresponding symbols and mentioned the conversion in the manuscript.

(6) l.113: Here, you should also mention the cooling by upwelling in the tropics.

Response:We have added it to the manuscript.

(7) l.117/118: Please clarify! Did you use both resolved and parameterized terms?

Response:Yes, we use both resolved (the dynamical heating rates) and parameterized (the radiative and moist heating rates) terms which are provided by MERRA2 (Modeling, G., & Office, A. 2015).

Reference:

Modeling, G., & Office, A. (2015). MERRA2 tavgM_3d_tdt_Np: MERRA2 3d, assimilation, monthly mean, temperature tendencies (p-coord, 0.625x0.5l42), version 5.12.4, greenbelt, MD, USA: Goddard space flight center distributed active archive center (gsfc daac) accessed: January 2021. doi: 10.5067/VILT59HI2MOY

(8) It is not clear to me what the unit "dBmonth" means that is used in Figs.1, 2, and 8. For a temperature response the unit Kelvin should appear somewhere!

Response:Sorry, it should be $K^2$ month and we have corrected the unit in figures.

(9) l.145: The good agreement between ERA5 and GNSS RO could also be an effect of using ECMWF temperatures as input for the wetPrf retrieval.

Response:Yes, GNSS RO WetPrf data use low-resolution ECMWF ERA-Interim profiles as a background. However, the temperature profiles are further derived by the 1DVAR technique (Wee and Kuo 2015). We have compared the GNSS RO temperature data in the UTLS with ERA-Interim, ERA5 and MERRA2 reanalyses in our previous study (Shangguan et al. 2019). The results show that the GNSS RO temperature in the UTLS has the best agreement with ERA5, rather than ERA-Interim, which are as used as inputs for 1DVAR.

Reference:

Wee, T.-K. and Kuo, Y.-H.: Advanced stratospheric data processing of radio occultation with a variational combination for multifrequency GNSS signals, Journal of Geophysical Research Atmospheres, 119, 11,011–11,039, 2015

Shangguan, M., Wang, W., and Jin, S.: Variability of temperature and ozone in the upper troposphere and lower stratosphere from multisatellite observations and reanalysis data, Atmospheric Chemistry and Physics, 19, 6659–6679, 2019

(10) l.154/155: Here you state that the largest SAO signal would be over the global monsoon regions. I disagree with this statement!

Having a look at Fig.2, the strongest signal is seen at 35deg in both hemispheres, as well at places near the equator, and not in the monsoon regions!

The monsoon regions are located around 15N and 15S, see:

https://www.wcrp-climate.org/documents/monsoon_factsheet.pdf

or:

Geen, R., Bordoni, S., Battisti, D. S., and Hui, K: Monsoons, ITCZs, and the concept of the global monsoon, Reviews of Geophysics, 58, e2020RG000700. https://doi.org/10.1029/2020RG000700, 2020.

At latitudes of 15deg the opposite is found in Fig.2: over the monsoon regions, there are minima of SAO PSD, and the signal is mostly insignificant!

Particularly, for North America the maximum SAO PSD is found over the Great Lakes, and not over Central America. Also in Eurasia, the maximum is between 35N and 45N, away from the monsoon region.

Perhaps it makes sense to compare the locations of the SAO PSD maxima with the seasonal changes of the jet streams, but this is just a speculation.

Response:Thank you very much for your suggestion. We agree with the reviewer that the largest SAO signal is not over the monsoon regions. After a further check, we think the strong SAO signals over that regions are likely related to both moist and dynamical processes. As seen in Figs. R1 and R2, there are strong upward motion over south Asia but downward motion from north Africa to central Asia. The strong upwelling brings a large number of water vapor from the surface to the upper troposphere, which condenses there and heats the atmosphere at around 200 hPa. At the same time, the strong downwelling leads to dynamical heating over the regions from north Africa to central Asia. We have also checked the annual variation of the jet streams as the reviewer suggested (Fig.R3). The seasonal changes of U over the focused areas are weaker than in other regions, which implies that the strong SAO signals may not be related to the seasonal changes of the jet streams. We have modified the sentences in the manuscript carefully.

[Figure]

Figure R1. The July (a) and January (b) averaged water vapor mixing ratio in ppmv of 2005-2017 microwave limb sounder (MLS) at 215hPa.

[Figure]

Figure  R2 The July (a) and January(b) averaged vertical velocity in Pa/s of 2005-2017 ERA5 at 200hPa.

[Figure]

Fig. R3 Annual cycle of the zonal mean eastward wind (U) at 200hPa averaged around the Asia region with blue lines ( 25ºN-45ºN, 20ºE-100ºE) and global region with red lines (25ºN-45ºN) using MERRA2 data.

(11) Fig.4, Fig.6, and Fig.9: Please plot from Jan to Jan, and not from Jan to Dec, otherwise the seasonal variations do not look cyclic.

Further, the peak in Dec is not given much space so that it is easily overlooked.

Response:Thank you for your suggestions. We have renewed these figures.

(12) l.225-237: Can you briefly mention the mechanisms that lead to the SST-SAO, and whether there are global phase variations?

Response:Thank you for your suggestions. According to Yashayaev and Yveryaev 2001, the SST-SAO in tropical Pacific and the Atlantic oceans is related to semiannual cycle of solar radiation. The SST-SAO in the tropical Indian Ocean is related to the Indian monsoon (Hu et al. 2004): The strong solar radiation is related with less cloud warms the water in spring, which leads highest SST; After the onset of the Indian summer monsoon, the strong coastal upwelling and offshore advection of cold waters in response to strong monsoon winds cause the temperature drop; The stronger solar radiations warms the water again, which leads to the second highest SST during boreal autumn; The cold winter monsoon from Asian continent cause another falling phase of SST.

In the supplement, Figures S7a-c show the correlation coefficients between the tropical(5° S-5° N), SH/NH mid-latitude UTLS-SAO and the SST-SAO. Correlation coefficients between SH/NH midlatitudes UTLS-SAO and SST-SAO are all negative values while the coefficients are positive between topical UTLS-SAO and SST SAO. According to the result, there are global phase variations. We have added the corresponding content to the manuscript.

References

Yashayaev, I.M. and Zveryaev, I.I. (2001), Climate of the seasonal cycle in the North Pacific and the North Atlantic oceans. Int. J. Climatol., 21: 401-417. https://doi.org/10.1002/joc.585

Ruijin, Hu., et al. On the mechanism of the seasonal variability of SST in the tropical Indian Ocean. Adv. Atmos. Sci. 22, 451 (2005). https://doi.org/10.1007/BF02918758

(13) l.292/293: I do not agree with the statement that the SAO is strongest in the monsoon regions.

Response:We have deleted the sentence.

(14) l.299-301: This is not necessarily an effect of the monsoons. The summer thunderstorm season at midlatitudes would have the same effect.

 Response:Thanks, we have deleted the monsoons.

Technical corrections:

Affiliations:

1: Informaiton ->  Information

  Wuhan,China -> Wuhan, China

Response:Corrected.

2:  Science,China ->  Science, China

Wuhan,China -> Wuhan, China

Response:Corrected.

l.24: maximum ->  maxima

Response:Corrected.

l.54: significant test -> significance test

Response:Corrected.

l.54: diagnose the SAO -> diagnose whether the SAO

Response:Corrected.

l.62: 2008 and -> 2008, and

Response:Corrected.

l.72: whereas -> for this,

Response:Corrected.

l.79 significant test -> significance test

Response:Corrected.

l.82  Details information ->  Detailed information

Response:Corrected.

l.85 Both reanalysis -> Both reanalyses

Response:Corrected.

l.110: with R is -> with R =

Response:Corrected.

l.122  MEERA2 -> MERRA2

Response:Corrected.

l.156:

signal overlies on the -> found in the

indicate that -> indicates that

Response:The sentence has been deleted according to your suggestions.

l.158, l.160: than other regions -> than in other regions

Response:Corrected.

Fig.3: labeling of the panels does not match with the figure caption, labeling is a-c-b-d, and not a-b-c-d

Response:Corrected.

l.180: two hemispheres -> the two hemispheres

Response:Corrected.

l.185: the are -> there are

Response:Corrected.

l.187: to show in -> and shown in

Response:Corrected.

l.188: Over all, -> Overall,

Response:Corrected.

l.192: Jannuary-February -> January-February

Response:Corrected.

l.207: but with -> but for the

Response:Corrected.

caption of Fig.7:

at 200 -> at 200 hPa

eddy , heating -> eddy, heating

Response:Corrected.

l.226: Easter Pacific -> Eastern Pacific

Response:Corrected.

l.247: with the Control simulation. -> with respect to the Control simulation.

Response:Corrected.

l.256: Figures 8c -> Figure 8c

Response:Corrected.

l.257: Figures 8d -> Figure 8d

Response:Corrected.

l.258: Compares the magnitude -> if we compare the magnitude

Response:Corrected.

l.259: in three model simulations -> in the three model simulations

Response:Corrected.

l.272: to three -> to the three

Response:Corrected.

caption of Fig.9:

The averaged of the extracted SAO singals -> The average extracted SAO signals

(SH) mid-latitudes -> (SH) mid-latitudes (b)

The blue, red -> The red, blue

Response:Corrected.

l.335 ???

In the theory of filter amplifires -> On the theory of filter amplifiers

Response:Corrected.

l.349: doi and page range missing for reference Gelaro et al. (2017)

Response:Added.

l.353: doi and page range missing

Response:The doi is added and there is no page for the reference.

units are missing in Figs. S2 and S5

Response:Added.

in Fig.S5 Panel (c) is not addressed in the figure caption

Response:Added.

**Response to Anonymous RC #2 's Comments,**

This paper examines the semi-annual oscillation (SAO) of temperature in the upper troposphere and lower stratosphere (UTLS) using the data from satellite observations, two reanalyses, and model simulations. Furthermore, this paper investigates the thermodynamic components which affect the SAO and find the connection between SAO in the surface and SAO in the UTLS. Overall, this is an interesting and well written paper. The manuscript contains material that is of interest to ACP readers. However, there are a few issues that I would like to see addressed before it is suitable for publication.

Response:We would like to sincerely thank the referee for the constructive comments and suggestions which helps to improve the manuscript significantly. Following the reviewer's comments/suggestions, we have revised the manuscript substantially. Please note that the manuscript is also altered according to the other 1 reviewer's comments and suggestions and all figures have been revised. More details can be found in the point-by-point responses as shown below.

Specific comments:

• Figure 1, Figure 2, and Figure 8

It is really difficult to understand Figure 1, Figure 2, and Figure 8. I was confused the first time that I read the paragraphs regarding these three figures. The color shading in the figures seems to indicate some variable related to the time period from the unit 'dBmonth' in the figures. However, the texts related to these figures describe the color shading as the strength or magnitude of temperature SAO. The PSD method is briefly mentioned in the end of Section 1. Since several figures and a lot of the results in the manuscript are based on the PSD method, it might be worth introducing the PSD method in more detail and explaining the meaning of 'dBmonth' to guide

the readers. If possible, it might be better to connect 'dBmonth' with 'Kelvin'

unit.

Response:We apologize for the confusing information. The unit should be $K^2$ month and we have corrected the unit in the figures. We add more description in lines 85-86.

• L138-L139

I'm not sure how the result is consistent with the results from Fueglistaler et al. (2009)?

Response:As described in the manuscript, there is also no significant annual cycle in the tropical upper troposphere in Fueglistaler et al. (2009). However, our research mainly focuses on the SAO, and we decide to delete this sentence after careful consideration.

• L154-L160

The UTLS SAO is significant almost in the whole zonal band in the middle latitude. Monsoon might amplify the amplitude of the SAO inside of some monsoon regions, but it does not suggest the direct connection between monsoon and the UTLS SAO outside of monsoon regions from Figure 2. Especially, the authors use Asian summer monsoon as an example to prove the influence of the transportation of water vapor and its condensation on the SAO. Does this also apply in the connection between Australian region and SAO? Maybe the authors should clarify the statement.

Response:Thank you very much for your suggestion. We have modified the relevant content about the monsoon. A significant part in the SH represented by MERRA2 model cannot be explained by ANA, dynamical and moist process, which indicates that the budget for the SH is not as reliable as the NH. Therefore, we use the energy budget of the simulation to analyze the Australian region. As shown in Fig. R4, the moist related heating rates are higher over the Australian region than other regions in the same latitude, indicating that the transportation of water vapor and its condensation also influences the SAO in the southern hemisphere. At the same time, the dynamical heating rates are also larger over the Australian region than other regions in the same latitude. Therefore, the more significant SAO signal over the Australian region is related to both moist and dynamical processes.

[Figure]

Fig. R4 Annual cycle of the zonal mean temperature (T), total heating rates (Total), dynamical heating rates (Dynamical), radiative heating rates (Radiative), and moist heating rates (Moist) at 200hPa averaged around the Australian region with blue lines ( 45°S-30°S, 30°E-180°E) and global region with red lines (25°N-45°N) using MERRA2 data. The total heating rates equal the sum of analyzed, dynamical, radiative, and mois theating rates.

• The authors mentioned "We then mainly focus on the SAO in the UTLS region (250-175hPa) in mid-latitudes (22.5 °-42.5°) hereafter in this study." in L142-143. However, almost all the results are about the region in 32.5 °-42.5° since Section 3.2. An explanation is required for taking different regions.

Response: Sorry for the mistake. We mainly focus on the region 32.5 °-42.5°. We have corrected the sentence.

• L207-L209 and Figure 6

There is no new information related to Figure 6 described in the manuscript. I would suggest to remove Figure 6 here and put it in the supplement.

Response:Thank you for your suggestion. We put Figure 6 to the supplement in the revised manuscript.

• L219

"by the -vTy term", should it be "by the eddy term"? Please check

Response:Yes, it should be "by the eddy term". We have corrected it.

• L236-L238

The nearly concurrent UTLS-SAO in the two hemisphere can explain the similar correlation patterns between SH and NH mid-latitude with SSTs, but not between SH/NH mid-latitude and tropical UTLS-SAO with SSTs.

Response:Yes, the tropical UTLS-SAO and SH/NH mid-latitude UTLS-SAO is out phase (Fig. R5). We correct the sentence.

[Figure]

Fig. R5 Time series of ERA5 zonal mean temperature at 200hPa averaged around the SM(32.5 °S-42.5°S, 20 °E-100 °E), NM(32.5 °N-42.5°N, 20 °E-100°E) and tropics (5 °S-5°N, 20 °E-100°E).

• L248-L251

"However, while the...in extra-tropics is important.", I would suggest to change

this sentence as "While the SST-SAO is removed globally, the SAO in the tropical upper-middle troposphere (500-175 hPa) is not significant. However, if the SST-SAO is removed only in the tropics, the upper troposphere SAO in the tropics is still significant, which indicates that the upper-middle troposphere SAO in the tropics is significantly influenced by the SST-SAO in the extra-tropics".

Response:Thank you very much for your suggestion. We have changed the sentence.

• L257-L258

"Such reduction of..." Maybe the reduction of the absolute magnitude is more

evident in the SH/NH compared to that in the tropics. The relative reduction seems comparable given the weak SAO in tropics in the control simulation.

Response:We further checked the relative difference between different model simulations (Fig. R6). The relative reduction is weaker in the tropics compared to those in SH/NH in the focusing region (250-175 hPa).

[Figure]

Fig. R6 (a) The relative difference of SAO PSD in % between the TPrmSAO and the Control simulations (TPrmSAO - Control) /Control for the period 2001-2017. (b) Same as (a), but for the relative difference between the rmSAO and the Control simulations (rmSAO - Control)/Control in %. The black dots mark area with significant

difference of the two time series at 95% level based on t-test. The dashed black lines mark the tropopause height calculated with GNSS RO data.

• L258-L260

This sentence just repeats the conclusion in L255-L256. Additionally, it should be "Figure1d, 8a-b"

Response:Thank you for your reminder. We delete this sentence.

• L281-L282 and L308

The modification in the summer moist heating is really small compared to that in the dynamical heating and radiative cooling for SH. I am not sure it is appropriate to make this conclusion for SH.

Response:Yes, the moist heating is really small compared to that in the dynamical heating and radiative cooling, but it shows good agreement with SAO. After careful consideration, we decide to add dynamical heating in these sentences. The modification should be related to moist heating , radiative cooling and dynamical heating.

Technical comments:

• L8

"the winter time ... the summer time maximum"–> "the austral winter time ... the austral summer time maximum"

To avoid the confusion, please check and add "austral" in front of winter and summer for SH in the whole manuscript

Response:Thank you for your suggestions. We have added them in the manuscript.

• L82

Details information –> Detailed information

Response:Corrected.

• L185

Therefore, the are –> Therefore, there are

Response:Corrected.

• L187

in Figure 5 –> Figure 5a

Response:Corrected.

• L190

Figure S4 –> Figure S4a

Response:Corrected.

• L216

Abalos et al. (2013) –> (Abalos et al., 2013)

Response:Corrected.

• L231

"between the tropical(5°S-5°N), SH/NH mid-latitude UTLS-SAO and the SST-SAO" might be better to change as "between the SST-SAO and the UTLS-SAO over the tropical(5°S-5°N) and the SH/NH mid-latitude"

Response:Corrected.

• L236

Pacific, whereas the SST-SAO are most pronounced (Figure S6) –> Pacific (Figure S6a), whereas the SST-SAO are most pronounced (Figure S5a)

Response:Corrected.

• L256

31% for NH and 55% for SH –> 31% for NHM and 55% for SHM

Please check and replace NH/SH with NHM/SHM through the whole manuscript when it is necessary, and add abbreviation about NHM/SHM in proper place ahead

Response:Thank you for your suggestions. We have checked them and corrected them in the manuscript.

• L263

in Figure 9 –> in Figure 9a

Response:Corrected.

• L267

SH (Figure 9b) –> SH (Figure 9b) compared to that in the NHM

Response:Corrected.

• L286-L287

except for a relatively large difference with other data sets in the tropical region –> except in the tropical region

Response:Corrected.

---

## Referee Report (RR1)

Review of the paper: "The semi-annual oscillation (SAO) in the upper troposphere and lower stratosphere (UTLS)" written by Shangguan and Wang,

**1 General comments:**

Thank the authors revising the manuscript based on the comments addressed before. Most of the comments are answered properly. However, there are a few minor points that I would like to see clarified before the manuscript is published.

**2 Specific comments:**

– L140-L141: The UTLS SAO signal is significant in the regions of 22.5°-42.5° in both hemispheres. The results are based on the regions of 32.5°-42.5°. Some connection needs to be built before L148 "We then mainly focus on the SAO..."

– L162-L166: Figure S3 and S4 do not help to explain the strong SAO signal over Asia and Australia. Compared to the regions with strong SAO signal in Figure 2, the regions with larger water vapor and strong upwelling are shifted toward the equator, where the SAO signal is weak. Perhaps the water vapor anomaly with zonal mean removed might show better agreement.

– L267-L268: "Such reduction of ...... than tropics." The SAO in tropics is located in the altitude between 400 hPa and 225 hPa. Figure R6 shows that the relative difference of SAO PSD in tropics (400-225 hPa) is larger than that in SHM/NHM UTLS (250-175 hPa). It might be better to change as "Such reduction of SAO PSD caused by removing SST-SAO is larger in the SHM and NHM than that in the tropics"

– L284-L285: "In October, the negative values ......reduced radiative cooling." The total heating rate is reduced only for the sensitivity simulation of rmSAO in October (Figure 9a)

– L289: In winter –> In austral winter

– L289-L290: "In April, the negative values......reduced radiative cooling." The total heating rate is reduced only for the sensitivity simulation of rmSAO-TP in April (Figure 9b)

– L306: in summer –> in austral summer or in February

– L307-L308: "The further energy ...... in the summer season" This conclusion applies in the NHM SAO. The peak of temperature for SHM in summer is not stronger than that in winter (Figure 3d and Figure 8b). It might be better to summarize it more precisely.

– Figure 4 caption: "The red, blue, dashed blue ...... analysis processes, respectively." –> The red, blue, dashed blue ...... analysis processes, dynamics, radiation, and condensation, respectively."

---

## Author Response (AR2)

**Response to Anonymous Referee #1 's Comments,**

1 General comments:

Thank the authors revising the manuscript based on the comments addressed before. Most of the comments are answered properly. However, there are a few minor points that I would like to see clarified before the manuscript is published.

We would like to sincerely thank the referee for the constructive comments and suggestions. Following the reviewer's suggestion, we have substantially revised the manuscript. Please note that the manuscript is also altered according to the other reviewer's comments and suggestions. More details can be found in the point-by-point responses as shown below.

2 Specific comments:

– L140-L141: The UTLS SAO signal is significant in the regions of 22.5°-42.5° in both hemispheres. The results are based on the regions of 32.5°-42.5° Some connection needs to be built before L148 "We then mainly focus on the SAO..."

Thank you very much for the very helpful comment. We haved added the connection before the sentence.

– L162-L166: Figure S3 and S4 do not help to explain the strong SAO signal over Asia and Australia. Compared to the regions with strong SAO signal in Figure 2, the regions with larger water vapor and strong upwelling are shifted toward the equator, where the SAO signal is weak. Perhaps the water vapor anomaly with zonal mean removed might show better agreement.

Thank you very much. We have made the water vapor anomaly with zonal mean removed as new Figure S3 in the supplement.

– L267-L268: "Such reduction of ...... than tropics." The SAO in tropics is located in the altitude between 400 hPa and 225 hPa. Figure R6 shows that the relative difference of SAO PSD in tropics (400-225 hPa) is larger than that in SHM/NHM UTLS (250-175 hPa). It might be better to change as "Such reduction of SAO PSD caused by removing SST-SAO is larger in the SHM and NHM than that in the tropics"

Thank you very much. We have corrected the sentence.

– L284-L285: "In October, the negative values ......reduced radiative cooling." The total heating rate is reduced only for the sensitivity simulation of rmSAO in October (Figure 9a)

Thank you for your suggestions. We have modified the sentence.

– L289: In winter –> In austral winter

Corrected.

– L289-L290: "In April, the negative values......reduced radiative cooling." The total heating rate is reduced only for the sensitivity simulation of rmSAO-TP in April (Figure 9b)

Thank you very much for the reminder. We have changed the sentence in the manuscript.

– L306: in summer –> in austral summer or in February

Corrected.

– L307-L308: "The further energy ...... in the summer season" This conclusion applies in the NHM SAO. The peak of temperature for SHM in summer is not stronger than that in winter (Figure 3d and Figure 8b). It might be better to summarize it more precisely.

Yes, we have added the NHM in the sentence.

**Response to Anonymous Referee #2 's Comments,**

In the revised paper "The semi-annual oscillation (SAO) in the upper troposphere and lower stratosphere (UTLS)" by Shangguan and Wang my suggested changes were satisfactorily addressed. The paper is now almost ready for publication in ACP. There are just a few remaining very minor comments as detailed below.

We would like to sincerely thank the referee for the constructive comments and suggestions which helps to improve the manuscript significantly. Following the reviewer's comments/suggestions, we have revised the manuscript substantially. Please note that the manuscript is also altered according to the other 1 reviewer's comments and suggestions. More details can be found in the point-by-point responses as shown below.

l.87: please include the ERA5 reference paper Hersbach et al. (2020)

Hersbach, H., Bell, B., Berrisford, P., Hirahara, S., Horanyi, A., Munoz-Sabater, J., Nicolas, J., Peubey, C., Radu, R., Schepers, D., Simmons, A., Soci, C., Abdalla, S., Abellan, X., Balsamo, G., Bechtold, P., Biavati, G., Bidlot, J., Bonavita, M., de Chiara, G., Dahlgren, P., Dee, D., Diamantakis, M., Dragani, R., Flemming, J., Forbes, R., Fuentes, M., Geer, A., Haimberger, L., Healy, S., Hogan, R. J., Holm, E., Janiskova, M., Keeley, S., Laloyaux, P., Lopez, P., Lupu, C., Radnoti, G., de Rosnay, P., Rozum, I., Vamborg, F., Villaume, S., and Thepaut, J.-N.: The ERA5 Global Reanalysis, Q. J. Roy. Meteor. Soc., 146, 1999-2049, https://doi.org/10.1002/qj.3803, 2020.

Thank you for your suggestion. We have added the reference in the manuscript.

The caption of Fig. S4 is not correct. It should address vertical velocity rather than water vapor.

Thank you very much for your suggestion. We have modified the caption of Fig. S4.

Caption of Fig.S9: panel (c) is not introduced

Thank you for the reminder. We have added it in the caption of Fig. S9.

---

## Author Response (AR3)

Response to Editor's Comments,

Dear Authors,

Thank you for addressing the reviewers' comments. Reading the last version of the paper, I noticed an inconsistency between Figures 8 and 9: while the extracted SAOs are in phase between the different WACCM simulations on Fig. 8, there seems to be a clear phase shift between rmSAO-TP and the others on Fig. 9. This discrepancy would have implications for the conclusions (last sentence of the abstract). Hence the paper can only be considered for publication after this difference has been explained or corrected.

We would like to sincerely thank the editor for the very helpful comments and suggestions. Following the editor's suggestion, we have substantially revised the manuscript. After a careful check of the figures, we found that the phase shift is due to a mistake in the date for the lines with the rmSAO simulation in Fig. 9. We are very sorry for the mistake. We have corrected Fig.9 and the corresponding descriptions for Fig.9 in the manuscript. Thanks again for your kind reminder.

I also have a few additional comments, which I would like you to take into account. The lines and pages refer to the tracked-change version of the manuscript.

P1 l 10 : 'Model simulations indicate that the SAO in the UTLS is partly modified by the SAO of sea surface temperatures (SSTs)' : if Figure 8 is correct, I would emphasize that 'Model simulations indicate that the SAO in the UTLS is only weakly affected by the existence of an SAO in sea surface temperatures (SSTs).' or something along those lines

Thank you for your suggestions. According to Fig. 7 and Fig.8, the SAO in the UTLS is partly affected by the existence of an SAO in SSTs in the SHM but weakly affected by the SAO in SSTs in the NHM. We have modified the sentence to summarize it more precisely.

p 1 l 20 : 'temperature variations' → 'temperature variability'

Corrected.

p2 l50 : ' both in' → 'both over'

Corrected.

p2 l 57-58 : 'an artifact e.g., just a harmonic part of the seasonal cycle. The

SAO variations investigated in this study are more sinus-like.' → 'an artifact. In

particular, the SAO investigated in this study is sinus-like with periods of ~ 6

months, i.e. it is not just a harmonic of the annual cycle'.

Thank you very much for your suggestion. We have renewed the sentence in the manuscript.

P3 l 80 : ' missing data in the early part of the GNSS RO record' which time

period has missing data ?

The GNSS RO data is available from May 2001. A few missing data are found during August-September 2001 and January-February 2002. We have added such information in the revised manuscript.

P6 l158 'data limitation': please specify what limitation (GNSS RO availability if

I am correct)

As we described in the Section 2.1 GNSS RO temperature data "CHAMP provides ca. 150 occulation evens globally per day from May 2001 to October 2008, and COSMIC began providing 1000-3000 occulation events per day since

late 2006", the observations of CHAMP is limited. We have modifed the sentence.

p9 l 260 'While' → 'When'

Corrected.

p 19-20 : Figures 4-5 : maybe replace 'analysis' and 'analysis process' by

'analysis tendency' for clarity

Corrected.